# Molecular and Functional Analysis of U-box E3 Ubiquitin Ligase Gene Family in Rice (*Oryza sativa*)

**DOI:** 10.3390/ijms222112088

**Published:** 2021-11-08

**Authors:** Me-Sun Kim, Kwon-Kyoo Kang, Yong-Gu Cho

**Affiliations:** 1Department of Crop Science, College of Agriculture and Life & Environment Sciences, Chungbuk National University, Cheongju 28644, Korea; kimms0121@cbnu.ac.kr; 2Division of Horticultural Biotechnology, Hankyong National University, Anseong 17579, Korea; kykang@hknu.ac.kr

**Keywords:** ubiquitination, E3 ligase, *OsPUB*, genome-wide, transcriptome, gene expression, cis-element, rice

## Abstract

Proteins encoded by U-box type ubiquitin ligase (PUB) genes in rice are known to play an important role in plant responses to abiotic and biotic stresses. Functional analysis has revealed a detailed molecular mechanism involving PUB proteins in relation to abiotic and biotic stresses. In this study, characteristics of 77 *OsPUB* genes in rice were identified. Systematic and comprehensive analyses of the *OsPUB* gene family were then performed, including analysis of conserved domains, phylogenetic relationships, gene structure, chromosome location, cis-acting elements, and expression patterns. Through transcriptome analysis, we confirmed that 16 *OsPUB* genes show similar expression patterns in drought stress and blast infection response pathways. Numerous cis-acting elements were found in promoter sequences of 16 *OsPUB* genes, indicating that the *OsPUB* genes might be involved in complex regulatory networks to control hormones, stress responses, and cellular development. We performed qRT-PCR on 16 *OsPUB* genes under drought stress and blast infection to further identify the reliability of transcriptome and cis-element analysis data. It was confirmed that the expression pattern was similar to RNA-sequencing analysis results. The transcription of *OsPUB* under various stress conditions indicates that the PUB gene might have various functions in the responses of rice to abiotic and biotic stresses. Taken together, these results indicate that the genome-wide analysis of *OsPUB* genes can provide a solid basis for the functional analysis of U-box E3 ubiquitin ligase genes. The molecular information of the U-box E3 ubiquitin ligase gene family in rice, including gene expression patterns and cis-acting regulatory elements, could be useful for future crop breeding programs by genome editing.

## 1. Introduction

Protein degradation regulated by the ubiquitin-proteasome system plays an important role in plants. This system is used to selectively degrade proteins through a process of poly-ubiquitination in eukaryotes [1,2]. In plants, adaptation in response to stress, signal transduction, DNA repair, differentiation, and immune response can be achieved through ubiquitination with the resulting degradation of components specific to these mechanisms [3,4]. Ubiquitination of a target protein is carried out involving a three-step (E1 → E2 → E3) conjugation cascade that detects specific ubiquitin signals. First, ubiquitins (Ub) are activated by a Ub-activating enzyme (E1). Ub is then transferred to a Ub-conjugating enzyme (E2). Finally, Ub is attached to the substrate protein by a specific Ub ligase enzyme (E3). Ub-conjugated proteins then become visible to the 26S proteasome, which is responsible for the protein degradation process [5,6].

E3 ubiquitin ligases are key factors in determining substrate specificity. They are classified into three main subfamilies: HECT (Homologous to E6-associated protein C-Terminus), RING (Real Interesting New Gene), and U-box according to the subunit construction and mechanism [7]. HECT proteins are single polypeptides that can form ubiquitin and thioester intermediates before target ubiquitination [8]. RING and U-box proteins are structurally related single polypeptides that use zinc chelation and hydrogen bonds and salt bridges to transfer ubiquitin from the E2-ubiquitin intermediate to the substrate, respectively [9,10].

U-box proteins are formed by a signature U-box domain containing 70 amino acids and a modified RING finger domain, which lacks characteristic zinc-chelating cysteine and histidine residues of the latter [11,12,13]. U-box proteins in plants (PUB) are classified into 13 groups based on the presence and structure of their domains. Most PUB genes contain armadillo repeats (ARM) that can mediate substrate interactions. Other protein domains include the U-box N-terminal domain (UND), Ser/Thr kinase domain (kinase), WD40 repeat (WD40), MIF4G domain (MIF4G), tetratricopeptide repeat (TPR), and peptidyl-prolyl isomerase (PPIase) [14,15,16]. Deletions of the PUB gene domain and point mutations of conserved amino acids within this domain can affect E3 ligase activity [17].

Plants use the ubiquitin/proteasome system (UPS) to promote changes in cellular protein content needed for continuous growth, development, and survival in a constantly changing environment. The ubiquitination of regulatory proteins such as transcription factors might be promoted or inhibited in response to environmental stimuli, leading to increased degradation or stabilization of these proteins, changes in gene expression patterns, and necessary cellular responses [18,19]. Targeted control of protein abundance can induce plants to respond appropriately by regulating signaling events, thereby reducing damage caused by external stress factors. A plant’s ability to effectively endure environmental stress is transformed when the UPS is interrupted. Mutations in regulatory particles of the 26S proteasome can reduce abiotic stress tolerance [20,21].

In Arabidopsis, RPN10 mutants are weakly resistant to high-salinity conditions, heat, and UV treatment [22]. Similarly, RPN1a mutants are sensitive to salinity and heat stress [23]. RPN12a and RPT2a mutants are also weakly resistant to heat stress [24]. Mutants of RPT2a or RPT5a render plants weakly resistant to the zinc deficiency condition [25]. The results of these mutation analyses indicate the importance of the 26S proteasome’s function in plant responses to adverse growth conditions. Studies on the importance of UPS functions in abiotic stress tolerance results from the analysis of ubiquitin and ubiquitin enzymes assay, and the expression of ubiquitin genes responds to stress [26,27]. Transgenic plants overexpressing mono- or poly-ubiquitin genes show strong tolerance to several abiotic stresses, including cold, salinity, and drought [28,29]. In addition, changes in ubiquitin protein expression can alter the response of plants to abiotic stresses. Expression levels of many E2 proteins are differentially regulated by abiotic stress. Expression levels of 14 of 39 E2 protein genes in rice have been shown to be upregulated or downregulated in response to drought and salt stresses [30]. The overexpression of GmUBC2 from soybean, VrUBC1 from mung bean, and AhUBC2 from dehydrated peanut plants can increase the tolerance of these plants to drought stress [31,32,33]. In addition, the increased expression of tobacco-derived NtUBC1 can promote cadmium resistance [34].

In rice, 77 U-box E3 ligase genes are known, of which only six genes have been reported functionally. That is, many rice U-box E3 genes have not been studied yet. Phenotypic interference due to functional duplication might be one of the reasons. Therefore, the objective of this study was to obtain information on the environmental response characteristics of individual PUB genes related to abiotic and biotic stresses tolerance in rice for breeding programs. Systematic characterization of the U-box E3 ubiquitin ligase gene family in rice was performed, including expression profiles, and cis-acting regulatory element structures of PUB genes related to stresses were analyzed. The results of this study will facilitate future crop breeding programs by genome editing of *OsPUB* genes.

## 2. Results

### 2.1. Characteristics and Chromosomal Distribution of U-box-Type E3 Ubiquitin Ligase Gene Family in Rice

According to recent reports on the U-box-type E3 ubiquitin ligase gene family in rice, 77 estimated PUB proteins were identified through a whole genome analysis algorithm and divided into eight classes according to domain configuration [35]. We adopted genomic and protein sequences of *OsPUB* genes from Gramene, RAP-DA, and NCBI. Identified rice protein sequences were then analyzed using ProtParam and the compute pI/Mw tool from ExPasy to further confirm physical parameters. Detailed characteristics of the *OsPUB* gene family are listed in Appendix A, including gene name, number of nucleotides, number of amino acids, molecular weight, theoretical isoelectric point, instability index, aliphatic index, grand average of hydropathicity (GRAVY), and chromosomal positions. These 77 OsPUB proteins have diverse molecular sizes in gene base pairs and amino acid lengths, ranging from 432 bp (OsPUB30) to 7675 bp (OsPUB27) with an average of 3598 bp in gene base pairs and from 212 a.a. (OsPUB68) to 1392 a.a. (OsPUB71) with an average of 605.13 amino acids. OsPUB68 had the smallest molecular weight (24.15 kDa) and OsPUB71 had the largest molecular weight (148.25 kDa). Their average molecular weight was 36 kDa. Theoretical isoelectric points (pIs) of these OsPUB proteins varied from 5.07 (OsPUB10) to 9.04 (OsPUB69) with an average of 6.82. Their aliphatic indices ranged from 69.29 (OsPUB26) to 109.68 (OsPUB31) with an average of 92.31. Among these 77 OsPUB proteins, 54 were hydrophobic proteins with a value of GRAVY less than 0. The remaining 23 genes were identified as hydrophilic proteins with a value of 0 or more.

Chromosomal locations of *OsPUB* genes were identified by extracting chromosomal data and mapping using the MapGene2Chromosome V2 program. The 77 *OsPUB* genes were distributed in 10 of 12 chromosomes of Oryza sativa, showing that each gene was unevenly distributed nonrandomly on the chromosome (Figure 1). Fifteen (the largest number) *OsPUB* genes were mapped on chromosome 2, while no *OsPUB* gene was mapped on chromosome 7 or 11. Fifteen out of 28 Class 2 *OsPUB* genes were distributed mainly on Chr 2, Chr 6, and Chr 8. The rest of the *OsPUB* genes were located on six chromosomes. Ten out of 16 Class 3 genes were mapped mainly on Chr 2 and Chr 4. Nine *OsPUB* genes in Class 4 were positioned on Chr 2 and Chr 10. The remaining seven *OsPUB* genes in Class 4 were distributed on Chr 3, 6, and 9. Genes belonging to Class 1, Class 5, Class 6, Class 7, and Class 8 were sporadically distributed on eight chromosomes.

### 2.2. Phylogenic Analysis and Gene Structure Analysis of OsPUB Genes

To better identify the phylogenetic relationship between PUB genes in rice and to evaluate the evolutionary history of the protein family, we performed multiple sequence alignment using ClustalW. Based on full-length sequences, a phylogenetic tree was constructed with MEGA 7.0 using the Neighbor-Joining method (Figure 2). As shown in the phylogenetic tree, all *OsPUB* genes were classified into three major subfamilies and one minor subfamily according to their structures or functions, consistent with previous studies on rice and Arabidopsis [35,36].

To understand exon/intron structures of individual *OsPUB* genes, their corresponding genomic DNA sequences were aligned using GSDS (Gene Structure Display Server). An unrooted tree categorized *OsPUB* genes into three major groups and one minor group with well-supporting bootstrap values, indicating that the classification was reliable (Figure 2 and Figure 3). In the Class 2 group, schematic structures revealed that most paralogs shared a similar exon/intron structure with one exon, while OsPUB2 and OsPUB71 had two exons. Class 2, the largest class, consisted of 32 genes with armadillo (ARM)/HEAT repeats. The ARM repeat is a double repeat sequence motif approximately 40 amino acids long. It was first identified in Drosophila melanogaster segment polarity gene armadillo [37]. It has been shown to be involved in protein–protein interactions [38]. The HEAT repeat derived its name from four different eukaryotic proteins, huntingtin, elongation factor 3, PR65/A subunit of protein phosphatase A, and TOR (target of rapamycin) [39]. Seventeen *OsPUB* genes in Class 3 shared a similar gene structure. Class 3, the second largest class of U-box proteins in rice, has a conserved domain containing ~100 amino acid residues located close to the C-terminus of the protein according to sequence alignment. In addition to a high proportion of leucine residues, a high percentage of homology and several highly conserved residues were detected [40]. This domain was named the GKL domain because it was rich in conserved glycine (G), lysine (K)/arginine (R), and leucine (L). Sequence alignments of Arabidopsis proteins of the same class were also found to exhibit very similar patterns. Arabidopsis thaliana U-box proteins and Nicotiana benthamiana proteins of this class were shown to be essential for plant defense and disease resistance [41]. Class 4 consisting of 19 U-box proteins has a kinase domain in the N-terminal region of the protein. Considering the involvement of phosphorylation in most cell processes, we speculate that members of this class of *OsPUB* genes might play a broad role in cells, if not all.

Conserved protein motifs of *OsPUB* genes were identified using MEME, a publicly available online tool. We identified 10 motifs with an amino acid sequence length that varied from 6 to 50. The motif sequence logo is presented in Figure 4. All OsPUB proteins except OsPUB1, OsPUB26, OsPUB64, and OsPUB69 contained motifs. The type and number of motifs were almost distinct for each family. Most member proteins in each family shared a common motif. A highly significant motif, motif 1 (YF[R]CPIS[L]E[V]MRDPVI[AA]TGQTYERE[A]I[E]RWL), showed regular expression. It is present in all protein sequences except OsPUB13, OsPUB18, and OsPUB73. Members of class 4 shared eight of 10 motifs, excluding motifs 8 and 9. Members of class 2 shared only five motifs. Members of class 3 shared only motifs 1, 2, and 7. OsPUB72 and OsPUB73 genes only shared a single motif.

### 2.3. Gene Expression Patterns of Rice U-box Gene Family Members under Abiotic and Biotic Stresses

To investigate expression changes of *OsPUB* genes under abiotic stress after treatment, we collected leaves at 0, 1, and 3 days after treatment (DAT) with drought stress and performed RNA-sequencing analysis. High-throughput sequencing produced a total of 167.3 million short reads, of which 98.82% (165.3 million) passed the quality control thresholds. To calculate the expression level and profile, clean reads were mapped onto the *Oryza sativa* reference genome using Trimmomatic v0.38. Thereafter, the read count of the transcript expression level was calculated using a StringTie v1.3.4d program. A comparative analysis was performed between samples (0 DAT, 1 DAT, and 3 DAT) based on the read count of StringTie calculated at the transcript level. In DESeq (V2.0), read count is normalized through the size factor and dispersion, and DEG is performed through the calculated log2 fold change value and FDR after normalization. DEGs were reported in log2FC with a corresponding *p*-value for each gene. Three samples were divided into three comparison groups. DEG analysis was then performed. In all three samples, the number of transcripts having expression values was 36,726. Values were selected according to statistical significance and differences in expression levels. As a result of a Heatmap for 77 *OsPUB* genes, it was confirmed that each gene showed differential expression. There were four expression pattern groups. For experimental samples comparing 14-day-old seedlings not treated with up- or downregulation to controls, 25 genes were found to be commonly upregulated and 16 were all downregulated under the given conditions above (Figure 5).

To investigate the putative role of the *OsPUB* gene in response to treatment with biotic stress, the specific expression pattern in rice during blast infection caused by fungus *Magnaporthe oryzae* was analyzed. Transcript accumulation patterns were investigated for each period (0 hpi, 12 hpi, and 24 hpi) of RNA-sequencing data of *Oryza sativa* L. ssp. *indica* after rice blast infection in GEO DataSets. Expression patterns at 0 hpi, 12 hpi, and 24 hpi were analyzed for 77 *OsPUB* genes out of 24,933 DEGs that were co-expressed at these three time points among all DEGs. Heatmap analysis for expression levels of all these genes showed that their expression profiles were quite diverse in response to rice blast infection. Five expression pattern groups were found. When the experimental sample was compared to the control, 23 and 14 genes were found to be commonly upregulated and downregulated, respectively, under the above conditions (Figure 6).

To further explore the role of *OsPUB* in regulating responses of rice to abiotic and biotic stresses, genes showing common expression patterns were identified. Results are shown in Table 1. Seven genes were identified to be upregulated and nine genes were found to be downregulated in common under both drought stress and rice blast infection. To identify protein structures of *OsPUB* genes showing common expression patterns, protein sequences were searched against the Simple Modular Architecture Research Tool (SMART) database to reveal distinct structures of these genes. Seven *OsPUB* genes commonly upregulated were found to have only a U-box domain in their protein structures. It was confirmed that the other 16 genes had a tetratrico peptide repeat region (TPR) and a protein kinase region in common. TPR is a structural motif present in a wide range of proteins. It mediates protein–protein interactions and the assembly of multi-protein complexes [42,43]. Protein kinases play a role in a multitude of cellular processes, including division, proliferation, apoptosis, and differentiation. Phosphorylation usually results in a functional change in the target protein by changing enzyme activity, cellular location, or association with other proteins [44,45].

### 2.4. Analysis of Cis-Acting Regulatory Elements in the Promoter of U-box E3 Ubiquitin Ligase Gene Family in Rice

To further explain the regulatory mechanism of the *OsPUB* gene in response to abiotic or biotic stress, we searched 3000 bp regions upstream of 16 *OsPUB* genes in the PlantCARE database as the expression level of genes was regulated by cis-regulating elements present in the promoter domain (Figure 7). Eighteen types of cis-acting regulatory elements were detected in five crops (*O. sativa*, *Z. mays*, *H. vulgare*, *T. aestivum*, and *S. tuberosum*). Lengths of these cis-elements varied from 5 bp to 14 bp. Cis-elements are grouped into different functional categories, as shown in Table 2. Five hormone-related elements (TATC-box, ABRE, CGTCA-motif, TGACG-motif, and P-box), seven stress-related elements (LTR, G-box, GT1-motif, Sp1, GATA-motif, I-box, and AT1-motif), and four cellular development-related elements (GCN4-motif, O2-box, GC-motif, and ARE) were classified into promoters of *OsPUB* genes. All 16 *OsPUB* genes contained 5 to 14 cis-elements related to stress or hormonal response or cellular development. ABRE (abscisic acid response) was detected in seven rice *OsPUB* genes. CGTCA-motif (MeJA-responsiveness) and TGACG-motif (MeJA-responsiveness) were detected in all 16 rice *OsPUB* genes. Different cis-acting regulatory elements related to different stress responses such as light and low temperature were observed in each of the five crops. The G-box element is known to be involved in light, ABA, MeJA, ethylene induction, and seed-specific expression [46]. It was found to be present in 14 of the 16 PUB genes except for *OsPUB28* and *OsPUB70*. Motifs involved in cellular development are relatively fewer in number compared to those involved in other responses. GCN4-motif is involved in endosperm expression. O2-box present in zein metabolism and circadian motifs plays a role in circadian control.

### 2.5. Gene Expression Levels of OsPUB Genes under Abiotic and Biotic Stresses

To further determine the responses of *OsPUB* genes to abiotic and biotic stresses, expression patterns of 77 *OsPUB* genes to drought stress and rice blast infection were examined using RNA-sequencing and in silico expression pattern analysis. Results revealed that *OsPUB* genes responded to both abiotic and biotic stresses, suggesting the authenticity of these genes and their potential role in stress response. In addition, quantitative real-time PCR was performed on 16 *OsPUB* genes, showing a common expression pattern to further confirm responses to abiotic and biotic stresses. Three biological replications were performed in all reactions. Results revealed that these genes were differentially expressed in leaves under normal conditions (0 HAT or no infection) (Figure 8) and under stress conditions (12 HAT, 24 HAT or 12 hpi, 24 hpi) (Figure 9). Interestingly, these 16 selected *OsPUB* genes showed a moderating expression pattern similar to RNA-sequencing or in silico expression analysis except for *OsPUB24* and *OsPUB57* under abiotic and biotic stresses. *OsPUB8, OsPUB37, OsPUB38, OsPUB41, OsPUB57, OsPUB66*, and *OsPUB67* showed low expression levels in rice leaves under normal conditions [35,36,47]. However, they showed upregulated expression levels during drought stress. Among them, *OsPUB38* and *OsPUB57* showed significantly upregulated expression levels (>2-fold). In addition, nine genes (*OsPUB21, OsPUB23, OsPUB24, OsPUB26, OsPUB28, OsPUB54, OsPUB56, OsPUB60*, and *OsPUB70*) were significantly downregulated during stress treatment. Among these genes, *OsPUB21* was confirmed to exhibit statistical significance between drought stress and rice blast infection expression analysis. The *N. benthamiana* CMPG1 gene homologous to the *Arabidopsis* PUB21 gene triggers cell death by the tomato receptor-like protein Cf-9 and the fungal elicitor Avr9 [41]. OsPUB57 revealed a higher expression in the resistant plants carrying the Pi9-resistant gene [48]. These results suggest that these selected genes of the *Oryza sativa* E3 ubiquitin ligase gene family are responsive to abiotic and biotic stresses, indicating potential roles of these genes in stress response.

## 3. Discussion

PUB genes in rice are known to play an important role in plant responses to abiotic and biotic stresses. Functional analysis revealed a detailed molecular mechanism involving PUB proteins in response to abiotic and biotic stresses [4,49,50]. To date, characteristics and functions of the PUB gene family have been identified in several plant species such as Arabidopsis [51], Soybean [52], Chinese cabbage [53], and Barley [54]. In this study, we performed a genome-wide analysis of PUB genes in rice by investigating their chromosome locations, phylogenetic relationships, gene structures, cis-acting elements, and gene expression profiles under various stress treatments. The classification of OsPUB proteins differs from those of other gene families based on additional domains besides the U-box, not based on genetic homology. Thus, in addition to the U-box domain, 77 OsPUB proteins can be grouped into three major groups and one minor group based on various other domains such as ARM/HEAT, GKL, TPR, and kinase. Mutations occurring at U-box motif positions often interfere with the structure and function of these enzymes [55,56]. Major PUB proteins whose biological functions have been identified belong to the PUB/ARM group [4]. The ARM repeat is a tandem repeat motif of about 40 amino acids in length that can bind to S-locus receptor kinase (SRK). It is an essential component of the signaling pathway downstream of SRK [38]. For example, PHOR1, a *Solanum tuberculosum* U-box protein with ARM repeats, appears to play a role in the signaling of gibberellin [57]. NtCMPG1, a *Nicotiana benthamiana* homologous with the Arabidopsis U-box protein AtPUB20/21, has been found to be essential for plant defense and disease resistance [41]. According to subsequent genetic structure and motif analysis, genes within the same subfamily were relatively preserved. However, properties of the *OsPUB* gene and its protein varied widely depending on different classifications. In addition to the U-box domain, 77 OsPUB proteins were found to be able to bind to different domains including ARM/HEAT, TPR domain, GLKs, and kinase. Eight members of PUB genes in rice only housed the domain. The TPR domain was found in *OsPUB67* and *OsPUB70.* WD40 repeats were found in *OsPUB 71* and *OsPUB72*. Therefore, PUB might be functionally diverse in plants. Analysis of the cis-acting regulatory elements in the promoter indicated that the PUB gene family was involved in stress-related mechanisms, hormone regulation, and cellular development. In our study, ABRE (abscisic acid response) was detected in seven *OsPUB* genes. However, CGTCA-motif (MeJA-responsiveness) and TGACG-motif (MeJA-responsiveness) were detected in putative promoter regions of all 16 rice *OsPUB* genes. This result indicates that the PUB gene might play a significant role during abiotic and biotic stresses, inducing ABA signal transduction and methyl jasmonate treatment in rice (Table 2). Based on RNA-Seq data under abiotic and biotic stress treatment conditions, expression levels of 16 *OsPUB* genes showing a common expression pattern were investigated using qRT-PCR. Interestingly, these 16 selected *OsPUB* genes showed a moderating expression pattern similar to RNA-sequencing or in silico expression analysis except for OsPUB24 and OsPUB57 under abiotic and biotic stresses. Among these genes, OsPUB21 was confirmed to exhibit statistical significance in expression analysis under both drought stress and rice blast infection conditions. In addition, OsPUB57 revealed a higher expression in the resistant plants carrying the Pi9-resistant gene [48]. These results suggest that these selected genes of the *Oryza sativa* E3 ubiquitin ligase gene family are responsive to abiotic and biotic stress treatments, indicating potential roles of these genes in stress response [28,29,58].

In summary, characteristics of 77 *OsPUB* genes in rice were identified. Systematic and comprehensive analyses of the *OsPUB* gene family were performed, including analysis of conserved domains, phylogenetic relationships, gene structures, chromosome locations, cis-acting elements, and expression patterns. Through transcriptome analysis, we confirmed that 16 *OsPUB* genes show similar expression patterns in drought stress and rice blast infection response pathways. Numerous cis-acting elements were found in promoter sequences of 16 *OsPUB* genes, indicating that the *OsPUB* genes might be involved in complex regulatory networks to control hormones, responses to stress, and cellular development. We performed qRT-PCR on 16 *OsPUB* genes under drought stress and rice blast infection to further identify the reliability of transcriptome and cis-element analysis data. It was confirmed that expression patterns were similar to RNA-sequencing analysis results except for 2 out of 16 genes. Transcriptome and qRT-PCR analyses indicated that sixteen *OsPUB* genes might play an important role in plant stress resistance mechanisms by showing expression patterns common to multiple stress response pathways. Taken together, these results indicate that genome-wide analysis of the *OsPUB* genes can provide a solid basis for functional analysis of rice E3 ubiquitin ligase genes. In addition, our study provides a theoretical basis for the future cultivation and breeding of crops suffering from environmental stress by characterizing and classifying U-box E3 ubiquitin ligase in rice, gene expression patterns, and cis-acting regulatory elements. The molecular information of the U-box E3 ubiquitin ligase gene family in rice, including gene expression patterns and cis-acting regulatory elements, could be useful for a future crop breeding program for producing stress-tolerant crops.

## 4. Materials and Methods

### 4.1. Identification of U-box E3 Ubiquitin Ligase Gene Family in Rice

All genomic and protein sequences of PUB genes of *Oryza sativa* were retrieved from Gramene (http://www.gramene.org accessed on 3 August 2020), Rice Genome Annotation Project (rice.plantbiolog.msu.edu accessed on 3 August 2020), and NCBI (http://www.ncbi.nlm.nih.gov/gene accessed on 3 August 2020). Molecular weight, pI (isoelectric point), instability index, aliphatic index, GRAVY (grand average of hydropathicity), number of amino acids, and number of nucleotides were analyzed using ProtParam and the compute pI/Mw tool from ExPasy (http://www.expasy.org/tools/ accessed on 3 August 2020). Chromosomal locations of the *OsPUB* genes family were mapped using the MapGene2Chromosome V2 program (http://mg2c.iask.in/mg2c_v2.0/ accessed on 7 May 2021).

### 4.2. Phylogenetic and Gene Strucure Analyses of OsPUB Gene Family

To identify and visualize the structural organization of the rice U-box E3 gene family, exon and intron structures of individual *OsPUB* genes were determined using GSDS (Gene structure display server; http://gsds.cbi.pku.edu.cn/ accessed on 20 November 2020) via the alignment of cDNAs with their corresponding genomic DNA sequences. The multiple sequence alignment of 77 *OsPUB* genes was performed using ClustalW. Results of the multiple sequence alignment were then used to construct a phylogenetic tree with MEGA version 7.0 (https://www.megasoftware.net/ accessed on 15 September 2021). Unrooted trees were constructed with the Neighbor-Joining method using the following parameters: Poisson correction, pairwise deletion, and 1000 bootstrap replicates. Conserved motifs in the rice U-box E3 gene family were identified using MEME suite (http://meme-suite.org/ accessed on 15 September 2021). A total of 10 motifs and a width limit of 200 amino acids with other default parameters were used for the analysis.

### 4.3. Prediction of Cis-Acting Elements in OsPUB Genes

PUB family gene promoters (3000 bp upstream of the translation start site) were obtained from the Rice Annotation Project Database (https://rapdb.dna.affrc.go.jp/index.html accessed on 23 August 2021). Stress response and hormone-related cis-acting regulatory elements in promoter sequences were surveyed using the PlantCARE database (http://bioinformatics.psb.ugent.be/webtools/plantcare/html accessed on 28 August 2021).

### 4.4. Plant Materials and Stress Treatment

For screening drought stress tolerance, four-week-old plants (*O. sativa* spp. *japonica* via Dongjin) in the greenhouse were transplanted into soils with a 5% moisture content. Screening was then performed without watering until the drought phenotype had appeared. The soil moisture content was measured using a Pro check device (Decagon devices, Pullman, WA, USA). Leaf samples were collected at 0, 1, and 3 days after treatment. Rice samples infected with *Magnaporthe oryzae* strains were collected from an experimental field of Chungbuk National University to screen gene expression against biological stress. Inoculated leaves with three biological replicates were harvested at 0 hpi, 12 hpi, and 24 hpi. The collected leaves were immediately frozen in liquid nitrogen and stored at −80 °C until RNA extraction. Total RNA was extracted from leaf tissues using the RNeasy Plant Mini Kit (QIAGEN, Germantown, MD, USA) according to the manufacturer’s instructions. The relative purity and concentration of RNA were estimated using the NanoDrop One (Thermo Fisher Scientific, Thermo Fisher Scientific, MA, USA) and checked on a 1.2% agarose gel. RNA samples were then stored in a −80 °C freezer.

### 4.5. cDNA Library Construction and RNA-Sequencing

Prior to library preparation, total RNAs of all samples were pooled for each experimental group. First-strand cDNAs were synthesized using the Oligo (dT)^20^ primer and ReverTra Ace^TM^ qPCR RT Master Mix (TOYOBO, Osaka, Japan). RNA-seq libraries for sequencing were prepared in triplicate and constructed according to the manufacturer’s protocol provided by Illumina Inc. (San Diego, CA, USA). Transcription profiles of rice seedlings were determined by RNA-seq on a DNACARE platform (http://www.dnacare.co.kr/web/ accessed on 16 December 2020) using Illumina HiSeq 2500 (Biomarker Technology Co., Rohnert Park, CA, USA), which generated raw data of 150-bp paired-end reads. RNA-seq reads were sorted after filtering adaptor sequences and low-quality bases. They were then mapped to the Os-Nipponbare-Reference IRGSP-1.0 reference genome (http://rapdb.dna.affrc.go.jp/download/archive/irgsp1/IRGSP-1.0_genome.fasta.gz accessed on 16 December 2020) using Trimmomatic v0.38. Complete library read alignment files were integrated from RAPDB (http://rapdb.dna.affrc.go.jp/download/archive/irgsp1 accessed on 16 December 2020) and MSU 7.0 databases (ftp://ftp.plantbiology.msu.edu/pub/data/Eukaryotic_Projects/o_sativa/annotation_dbs/pseudomolecules/version_7.0/all.dir/all.gff3 accessed on 16 December 2020) using HISAT2.

### 4.6. Differential Expression Analysis

Expression levels of genes were calculated and normalized to fragments per kilobase of transcript per killion fragments mapped (FPKM) to identify differentially expressed genes (DEGs). StringTie v1.3.4d is a software for estimating gene and transcript expression levels from RNA-Seq data. The measurement result of the transcript expression level through assembly was used for DEG analysis. DESeq is a tool for analyzing the difference in expression levels of RNA-Seq expression profiles, and it provides a normalization between comparative samples that minimizes the loss of biological significance by implementing various statistical methods. DEGs were first selected using Student’s *t*-test (*p* < 0.05) and log2FC (fold change) > 1 between before stress treatment and after stress treatment.

### 4.7. Expression Analysis of OsPUB Genes

For qRT-PCR, first-strand cDNAs were synthesized using the Oligo (dT)^20^ primer and ReverTra Ace^TM^ qPCR RT Master Mix (TOYOBO, Osaka, Japan). Primer sets were designed through primer3plus software (Appendix A). qRT-PCR was performed using iQ SYBR Green Supermix (Bio-Rad, Hercules, CA, USA) on a BioRad CFX96 Detection System (Bio-Rad Laboratories, Hercules, CA, USA) according to the manufacturer’s instructions. Primers for actin as an internal control were used to normalize the results of real-time qRT-PCR.

## Figures and Tables

**Figure 1 ijms-22-12088-f001:**
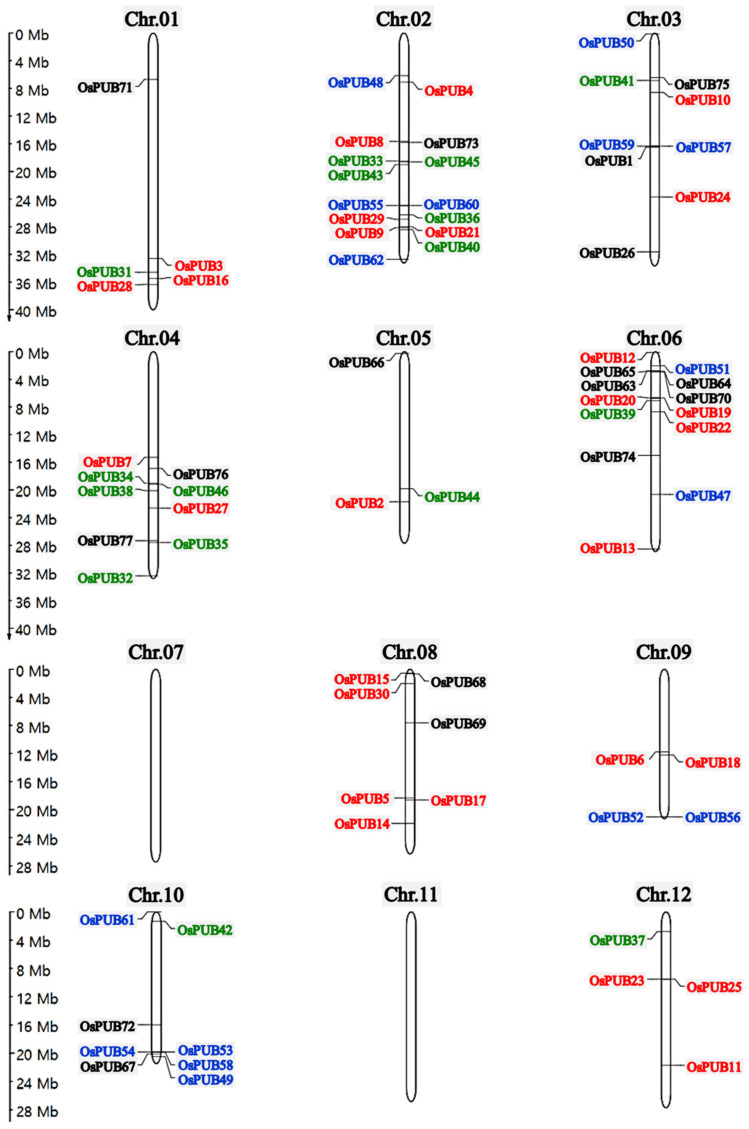
Chromosome mapping of U-box E3 ubiquitin ligase gene family in *Oryza sativa*. The rice U-box gene family is divided into different classes shown by different colors: red color, Class 2; green color, Class 3; blue color, Class 4; black color, Class 1, Class 5, Class 6, Class 7, and Class 8. Chromosome number is shown at the top of the bar. The scale bar on the left indicates chromosome length (Mb).

**Figure 2 ijms-22-12088-f002:**
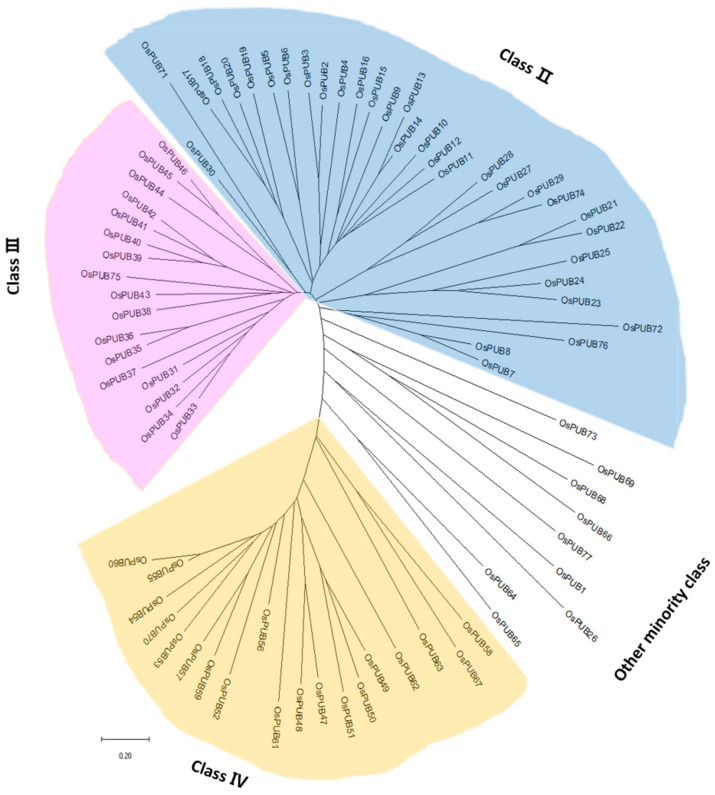
Phylogenetic tree of U-box E3 ubiquitin ligase gene family members in *Oryza sativa* constructed with the Neighbor-Joining method. The rice U-box gene family is grouped into different classes shown by different colors. Blue color represents Class 2. Pink color indicates Class 3. Yellow color represents Class 4. White color indicates Class 1, Class 5, Class 6, Class 7, and Class 8.

**Figure 3 ijms-22-12088-f003:**
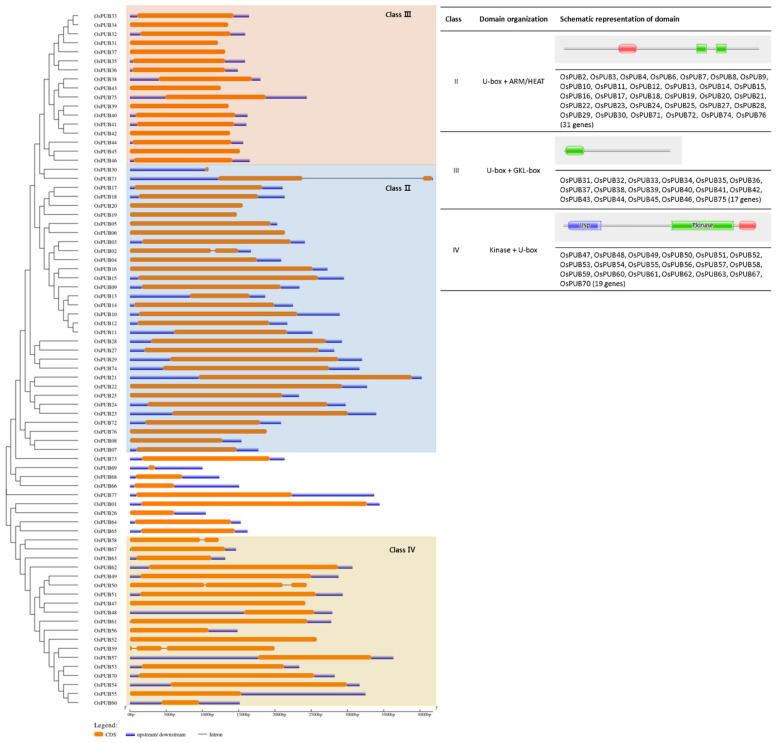
Exon/intron distribution of 77 rice U-box gene family members was analyzed with the GSDS tool. Coding sequences were compared with their corresponding genomic sequences. The orange box represents the CDS. The continuous black line represents the intron region. Blue boxes represent upstream/downstream regions.

**Figure 4 ijms-22-12088-f004:**
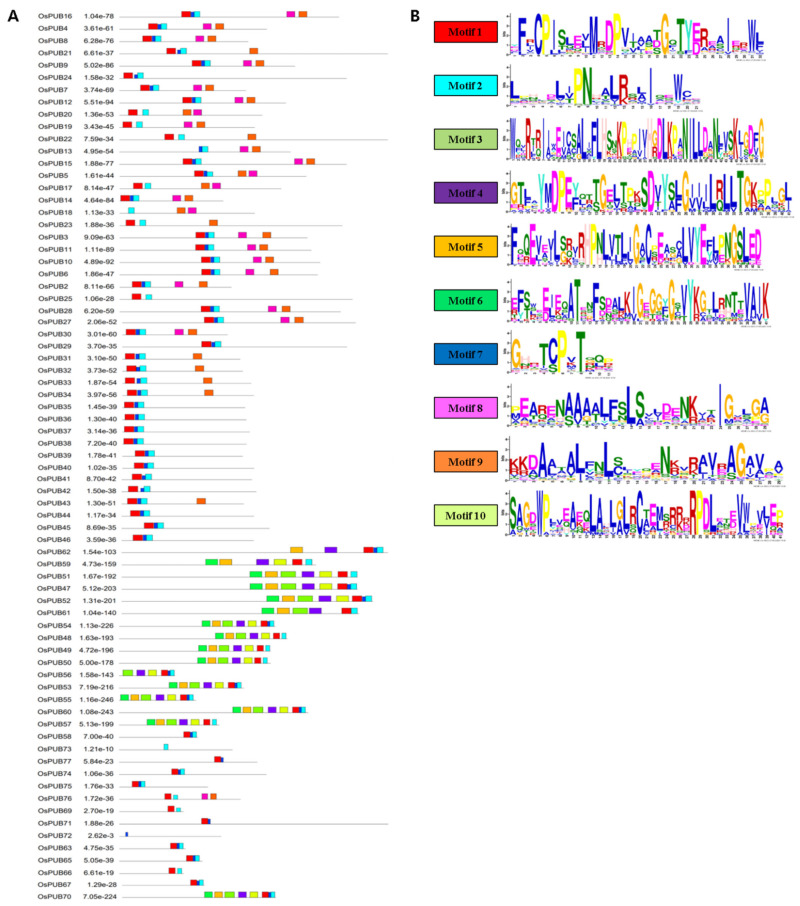
Gene structure and conserved protein sequence patterns of *OsPUB* gene family. (**A**) Conserved motifs in *OsPUB* genes. These motifs were identified with the MEME program. Ten conserved motifs are displayed with different colored boxes. (**B**) Sequence logos of 10 conserved motifs. The height of the letter showing amino acid residue at each position represents the degree of conservation. Numbers on the *x*-axis represent residue positions in motifs. The *y*-axis represents content measured in bits.

**Figure 5 ijms-22-12088-f005:**
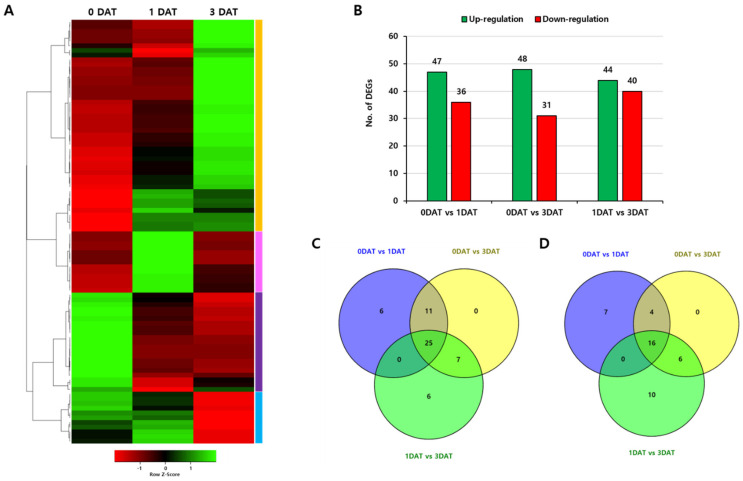
Expression patterns of 77 rice U-box gene family members in leaves under drought stress. (**A**) Heatmap showing expression of total differential expressed genes with increased or decreased trend under drought treatment conditions. (**B**) Total number of upregulated and downregulated genes. (**C**) Venn diagram of upregulated genes in the two samples vs. control. (**D**) Venn diagram of downregulated genes in the two samples vs. control. Two samples are 1 DAT and 3 DAT.

**Figure 6 ijms-22-12088-f006:**
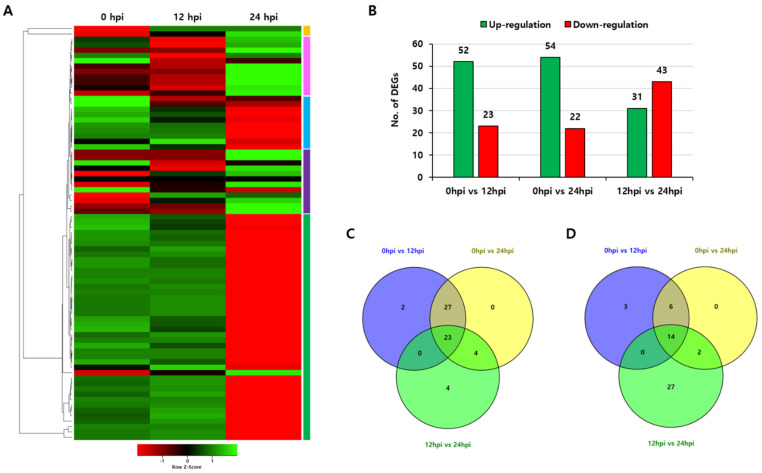
Expression patterns of 77 rice U-box gene family members in *indica* rice plants infected with *Magnaporthe oryzae*. (**A**) Heatmap showing expression of total differential expressed genes with increased or decreased trend. (**B**) Total number of upregulated and downregulated genes. (**C**) Venn diagram of upregulated genes in the two samples vs. control. (**D**) Venn diagram of downregulated genes in the two samples vs. control. Two samples are 12 hpi and 24 hpi.

**Figure 7 ijms-22-12088-f007:**
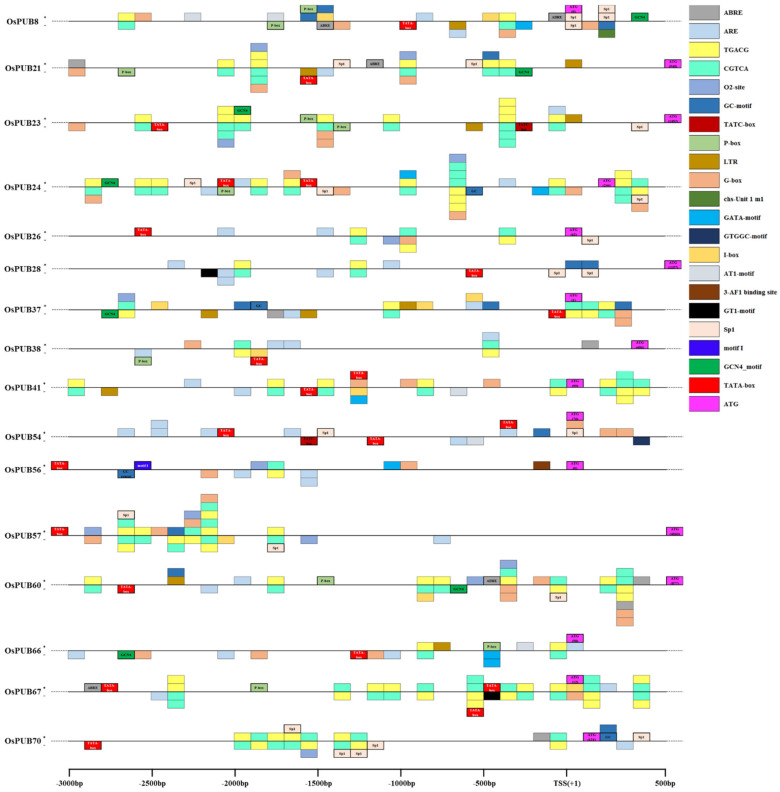
Distribution of stress-related cis-acting elements in promoter sequences of 16 *OsPUB* genes. Cis-elements distributed on the sense strand and the reverse strand are indicated above and below black lines, respectively. The 3 kb sequences upstream of the Transcription Start Site (TSS) of *OsPUB* genes can be estimated using the scale per 500 bp at the above. 3-AF1 binding site: light responsive element; ABRE: cis-acting element involved in the abscisic acid responsiveness; ARE: cis-acting regulatory element essential for the anaerobic induction; AT1-motif: part of a light responsive module; CGTCA-motif: cis-acting regulatory element involved in the MeJA-responsiveness; chu-Unit 1 m1: part of a light responsive element; GATA-motif: part of a light responsive element; G-box: cis-acting regulatory element involved in light responsiveness; GC-motif: enhancer-like element involved in anoxic specific inducibility; GT1-motif: light responsive element; I-box: part of a light responsive element; LTR: cis-acting element involved in low-temperature responsiveness; O2-site: cis-acting regulatory element involved in zein metabolism regulation; P-box: gibberellin-responsive element; Sp1: light responsive element; TATA-box: core promoter element around −30 of transcription start; TATC-box: core promoter element around −30 of transcription start; TGACG-motif: cis-acting regulatory element involved in MeJA-responsiveness. The cis-acting regulatory elements of *Oryza sativa* are indicated by a bold border and written inside a box. Cis-elements for other crops (*Zea mays*, *Hordeum vulgare*, *Triticum aestivum*, and *Solanum tuberosum*) are displayed only with thin borders. * The regions having several boxes (such as 1800 bp of OsPUB21) are with several cis-elements in a 100 bp interval. The closest box on the line means near to the starting bp of each multiple-box region.

**Figure 8 ijms-22-12088-f008:**
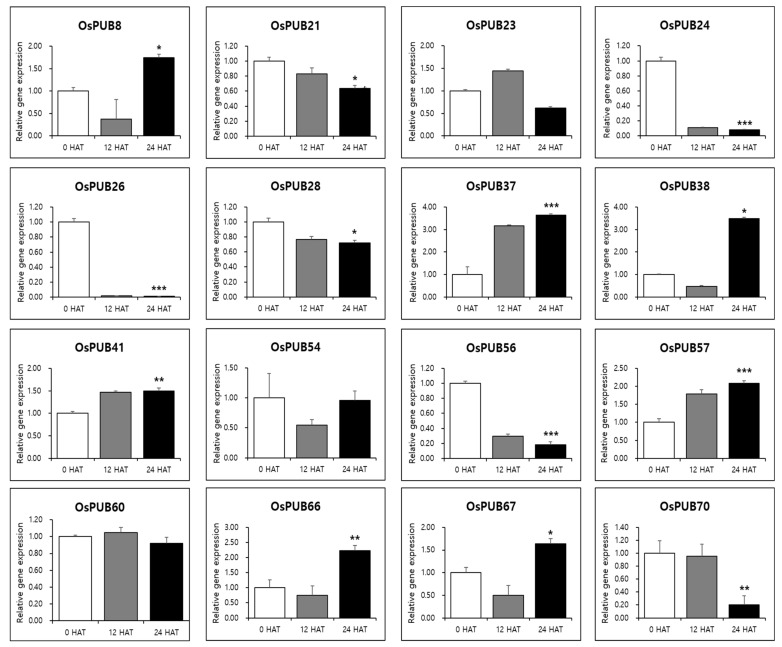
Expression patterns of *OsPUB* genes in response to drought stress treatment analyzed by qRT-PCR. Relative gene expression level was normalized against rice *Actin* gene. Values are presented as means ± standard error. Three independent biological repeats were performed. Bars represent standard deviations (SD) of three technical replicates. A significant difference is indicated by an asterisk according to *t*-test (* *p* < 0.05, ** *p* < 0.01, and *** *p* < 0.001).

**Figure 9 ijms-22-12088-f009:**
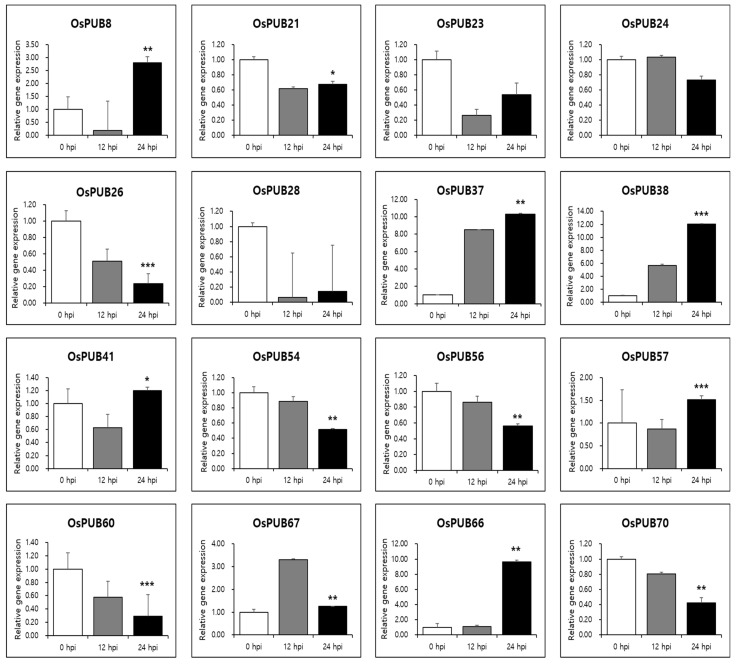
Expression patterns of *OsPUB* genes in response to infection of *Magnaporthe oryzae* analyzed by qRT-PCR. Relative gene expression was normalized against rice *Actin* gene. Values are presented as means ± standard error. Three independent biological repeats were performed. Bars represent standard deviations (SD) of three technical replicates. A significant difference is indicated by an asterisk according to *t*-test (* *p* < 0.05, ** *p* < 0.01, and *** *p* < 0.001).

**Table 1 ijms-22-12088-t001:** A list of *OsPUB* genes showing the same expression pattern under abiotic stress (drought condition) and biotic stress (blast infection).

No.	Gene Name	Locus No.	Class	Common Regulation	Drought Stress		Blast Infection	
					Log_2_FC (0 DAT vs. 1 DAT)	Log_2_FC (0 DAT vs. 3 DAT)	Log_2_FC (0 hpi vs. 12 hpi)	Log_2_FC (0 hpi vs. 24 hpi)
1	OsPUB08	LOC_Os02g28720	2	UP	1.223 **	2.485	0.208	0.209
2	OsPUB21	LOC_Os02g49520	2	DOWN	−3.078 **	−5.808 **	−1.501 *	−1.686 **
3	OsPUB23	LOC_Os12g17900	2	DOWN	−0.442	−0.694	−0.039	−0.136
4	OsPUB24	LOC_Os03g45420	2	UP	1.850 *	3.365 **	1.416	1.456 **
5	OsPUB26	LOC_Os03g60140	5	DOWN	−0.338	−1.248	−0.402	−0.679
6	OsPUB28	LOC_Os01g67500	2	DOWN	−0.071 **	−1.764 ***	−0.739 **	−0.752 **
7	OsPUB37	LOC_Os12g06410	3	UP	0.069 *	0.248 *	1.884 **	2.165 ***
8	OsPUB38	LOC_Os04g35680	3	UP	0.493	1.910 **	3.422	3.852
9	OsPUB41	LOC_Os03g13740	3	UP	2.724 *	3.576 **	3.578 **	3.723 *
10	OsPUB54	LOC_Os10g40100	4	DOWN	−8.452	−0.141	−0.686	−0.878
11	OsPUB56	LOC_Os09g39640	4	DOWN	−0.468	−1.457	−0.149	−0.228
12	OsPUB57	LOC_Os03g31070	4	DOWN	−1.264 *	−2.056 **	−0.406	−0.718 *
13	OsPUB60	LOC_Os02g44599	4	DOWN	−0.768	−2.068	−0.527	−0.567
14	OsPUB66	LOC_Os05g01460	7	UP	0.749	1.935	0.158	0.247
15	OsPUB67	LOC_Os10g40490	7	UP	3.412 **	4.337 **	1.478 **	1.531 **
16	OsPUB70	LOC_Os06g06760	8	DOWN	−3.998	−1.096	−0.104	−0.482

* Three independent biological repeats were performed. A significant difference is indicated by asterisks according to *t*-test (* *p* < 0.05, ** *p* < 0.01, *** *p* < 0.001).

**Table 2 ijms-22-12088-t002:** Category of cis-elements extracted from 3000 bp upstream region of *OsPUB* genes using PlantCARE.

Function	Cis-Element	Sequence	Presence of Cis-Element in PUB Genes	
			In PUB Genes	In Diffenrent Species
Cis-acting element involved in the abscisic acid responsiveness	ABRE	GCCGCGTGGC	OsPUB8, OsPUB21, OsPUB60, OsPUB67	*Oryza sativa*
		GACACGTGGC	OsPUB37, OsPUB70	*Triticum aestuvum*
		CGCACGTGTC	OsPUB21, OsPUB38, OsPUB60	*Hordeum vulgare*
Cis-acting regulatory element essential for the anaerobic induction	ARE	AACCA	OsPUB8, OsPUB21, OsPUB23, OsPUB24, OsPUB26, OsPUB28, OsPUB37, OsPUB38, OsPUB41, OsPUB54, OsPUB56, OsPUB57, OsPUB60, OsPUB66, OsPUB67, OsPUB70	*Zea mays*
Cis-acting regulatory element involved in the MeJA-responsiveness	TGACG	TGACG	OsPUB8, OsPUB21, OsPUB23, OsPUB24, OsPUB26, OsPUB28, OsPUB37, OsPUB38, OsPUB41, OsPUB54, OsPUB56, OsPUB57, OsPUB60, OsPUB66, OsPUB67, OsPUB70	*Hordeum vulgare*
	CGTCA	CGTCA		
Cis-acting regulatory element involved in zein metabolism regulation	O2-site	GATGACATGG	OsPUB37, OsPUB70	*Zea mays*
		GATGATGTGG	OsPUB21, OsPUB26, OsPUB56, OsPUB57	
		GTTGACGTGA	OsPUB21, OsPUB23, OsPUB24, OsPUB57, OsPUB60	
Enhancer-like element involved in anoxic specific inducibility	GC-motif	CCCCCG	OsPUB8, OsPUB21, OsPUB28, OsPUB37, OsPUB54, OsPUB57, OsPUB60, OsPUB70	*Zea mays*
Cis-acting element involved in gibberellin-responsiveness	TATC-box	TATCCCA	OsPUB23, OsPUB54	*Oryza sativa*
Gibberellin-responsive element	P-box	CCTTTTG	OSPUB8, OsPUB21, OsPUB23, OsPUB24, OsPUB38, OsPUB60, OsPUB66, OsPUB67	*Oryza sativa*
Cis-acting element involved in low-temperature responsiveness	LTR	CCGAAA	OsPUB8, OsPUB21, OsPUB23, OsPUB37, OsPUB41, OsPUB60, OsPUB66	*Hordeum vulgare*
Cis-acting regulatory element involved in light responsiveness	G-box	TCCACATGGCA	OsPUB41, OsPUB56	*Triticum aestivum*
		CACGTC	OsPUB8, OsPUB21, OsPUB23, OsPUB24, OsPUB26, OsPUB37, OsPUB38, OsPUB41, OsPUB54, OsPUB56, OsPUB57, OsPUB60, OsPUB66, OsPUB67	*Zea mays*
Part of a light responsive element	chs-Unit 1 m1	ACCTAACCCGC	OsPUB8	*Hordeum vulgare*
	GATA-motif	AAGGATAAGG	OsPUB8, OsPUB24, OsPUB41, OsPUB56, OsPUB66	*Solanum tuberosum*
	GTGGC-motif	CAGCGTGTGGC	OsPUB54	*Hordeum vulgare*
	I-box	AGATAAGG	OsPUB37, OsPUB57	*Triticum aestivum*
		gGATAAGGTG	OsPUB8, OsPUB37, OsPUB38, OsPUB41, OsPUB60, OsPUB67	*Zea mays*
	AT1-motif	AATTATTTTTTATT	OsPUB8, OsPUB41, OsPUB54, OsPUB66, OsPUB67	*Solanum tuberosum*
Light responsive element	3-AF1 binding site	TAAGAGAGGAA	OsPUB56	*Solanum tuberosum*
	GT1-motif	GTGTGTGAA	OsPUB28	
	Sp1	GGGCGG	OsPUB8, OsPUB21, OsPUB23, OsPUB24, OsPUB26, OsPUB28, OsPUB54, OsPUB57, OsPUB60, OsPUB70	*Oryza sativa*
Cis-regulatory element involved in endosperm expression	GCN4_motif	TGAGTCA	OsPUB8, OsPUB21, OsPUB23, OsPUB24, OsPUB37, OsPUB60, OsPUB66	*Oryza sativa*

## Data Availability

Not applicable.

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
