# Peer review of "Molecular and Functional Analysis of U-box E3 Ubiquitin Ligase Gene Family in Rice (Oryza sativa)"

_ijms, 2021, doi:10.3390/ijms222112088_

Round 1

Reviewer 1 Report

The manuscript reports interesting results about the characterization of U-box E3 ubiquitin ligase gene family in rice. In my opinion, the paragraphs relating to the phylogenetic analysis and the analysis of the structure of genes, including the characterization of the Cis-acting regulatory elements, was adequately detailed,  while I think it is necessary to integrate the paragraph relating to the results of the gene expression under abiotic and biotic stresses. Moreover, the discussions about the gene expression results (both for RNA-seq and qRT-PCR data) are in my opinion too synthetic, also few bibliographic references have been reported.

Furthermore, I did not find in the manuscript a reference to a public repository where the sequencing data was deposited, so we are able only to see differentially expressed genes but not the raw and all normalized data.

Line 110 Please, report in full the definition of "GRAVY"

Line 113 Change OsPUB63 to OsPUB68

Line 119 Please, change “a value of less than 0” to “a value of GRAVY less than 0”

Line 123 Italicize “Oryza sativa”

Line 126 Please, change “31” to “28”

Line 128 Change “19” to “16”

Line 129 Change “Chr 2 & 10” to “Chr 2 and Chr 10”

Line 153 I suggest you change “black color” to “white color”

Line 160 The phylogenetic tree reports 32 genes in Class II, so please change “31” to “32”

Line 162 Please, italicize “Drosophila melanogaster”

Line 171 Please, add “(L)” after “leucine”

Line 173 Add “thaliana” after “Arabidopsis” and italicize “Nicotiana benthamiana”

Line 199 Change “abitic” to “abiotic”

Line 316 It is not clear to me why only the OsPUB21 gene is mentioned, from the analysis of figures 8 and 9 other genes would appear to exhibit statistical significance between drought stress and rice blast infection expression analysis. In this regard, I do not understand why in figures 8 and 9 the significance was not indicated with asterisks in those cases in which there would seem to be a significant difference (for example OsPUB24, OsPUB26 ...)

Line 370 Why is only the OsPUB21 gene mentioned? In my opinion, there are other significantly differentially expressed genes that could be discussed

Line 380 Please, remove the hyphen in "elements"

Line 384 Change “transcriptiome” to “transcriptome”

Line 388 Remove the hyphen in “patterns”

Line 389 Change “OsPUB gene” to OsPUB genes”

Table 3 Please, change “Gebberellin-responsive element” to “Gibberellin-responsive element”

Table 3 Please, change “Cis-acting element involved in low-temperature resonsiveness” to “Cis-acting element involved in low-temperature responsiveness”

Line 434 Was the quality of the RNA samples evaluated only by the NanoDrop One?

Line 436 Please, change “cDAN” in “cDNA”

Line 438 Please, could you indicate which library preparation kit was used? and how was the quality of the cDNA libraries assessed?

Author Response

Responses to Reviewer 1

Title: Genome-wide analysis of biologically important cis-regulatory elements of E3 ubiquitin ligase gene family in rice (Oryza sativa L.)

Comments and Suggestions for Authors

The manuscript reports interesting results about the characterization of U-box E3 ubiquitin ligase gene family in rice. In my opinion, the paragraphs relating to the phylogenetic analysis and the analysis of the structure of genes, including the characterization of the Cis-acting regulatory elements, was adequately detailed, while I think it is necessary to integrate the paragraph relating to the results of the gene expression under abiotic and biotic stresses. Moreover, the discussions about the gene expression results (both for RNA-seq and qRT-PCR data) are in my opinion too synthetic, also few bibliographic references have been reported.

Furthermore, I did not find in the manuscript a reference to a public repository where the sequencing data was deposited, so we are able only to see differentially expressed genes but not the raw and all normalized data.

(Comment) I think it is necessary to integrate the paragraph relating to the results of the gene expression under abiotic and biotic stresses.

--- I am grateful for your critical comments. We have explained on gene expressions under abiotic and biotic stresses in Lines 310~334.

(Comment) Moreover, the discussions about the gene expression results (both for RNA-seq and qRT-PCR data) are in my opinion too synthetic, also few bibliographic references have been reported.

--- Thank you for your critical comments. We have included citations on OsPUB41, OsPUB57 and OsPUB67 in Line 323 and in Reference.

(Comment) Furthermore, I did not find in the manuscript a reference to a public repository where the sequencing data was deposited, so we are able only to see differentially expressed genes but not the raw and all normalized data.

--- I am grateful for your critical comments. We are storing the RNA sequence data in the personal DB. So we will be willing to provide the data sets if any scientist request to us.

Line 110 Please, report in full the definition of "GRAVY"

--- I am grateful for your critical comments. We revised it in Line 113 of the manuscript as follows: Grand average of hydropathicity (GRAVY)

Line 113 Change OsPUB63 to OsPUB68

--- I am grateful for your critical comments. We revised it in Line 117 of the manuscript as follows: OsPUB68

Line 119 Please, change “a value

--- I am grateful for your critical comments. We revised it in Line 122 of the manuscript as follows: a value of GRAVY less than 0.

Line 123 Italicize “Oryza sativa”

--- I am grateful for your detailed comments. We revised it in Line 127 of the manuscript.

Line 126 Please, change “31” to “28”

--- I am grateful for your critical comments. We revised it in Line 130 of the manuscript.

Line 128 Change “19” to “16”

--- I am grateful for your critical comments. We revised it in Line 132 of the manuscript.

Line 129 Change “Chr 2 & 10” to “Chr 2 and Chr 10”

--- I am grateful for your critical comments. We revised it in Line 133 of the manuscript.

Line 153 I suggest you change “black color” to “white color”

--- I am grateful for your critical comments. We revised it in Line 156 of the manuscript.

Line 160 The phylogenetic tree reports 32 genes in Class II, so please change “31” to “32”

--- I am grateful for your detailed comments. We revised it in Line 163 of the manuscript.

Line 162 Please, italicize “Drosophila melanogaster”

--- I am grateful for your critical comments. We revised it in Line 165 of the manuscript.

Line 171 Please, add “(L)” after “leucine”

--- I am grateful for your detailed comments. We revised it in Line 174 of the manuscript.

Line 173 Add “thaliana” after “Arabidopsis” and italicize “Nicotiana benthamiana”

--- I am grateful for your detailed comments. We revised it in Line 176 of the manuscript.

Line 199 Change “abitic” to “abiotic”

--- I am grateful for your detailed comments. We revised it in Line 204 of the manuscript.

Line 316, It is not clear to me why only the OsPUB21 gene is mentioned, from the analysis of figures 8 and 9 other genes would appear to exhibit statistical significance between drought stress and rice blast infection expression analysis. In this regard, I do not understand why in figures 8 and 9 the significance was not indicated with asterisks in those cases in which there would seem to be a significant difference (for example OsPUB24, OsPUB26 ...)

--- I am grateful for your detailed comments. We revised it in Line 329 of the manuscript as follows: We included more information as follows: OsPUB57 revealed a higher expression in the resistant plants carrying the Pi9-resistant gene [48] in Lines 331~332. Also, we described about OsPUB21 in the Discussion. We have included the asterisks in Figure 8 & 9. If no asterisk, no significance detected in Figure 8 & 9.

Line 370 Why is only the OsPUB21 gene mentioned? In my opinion, there are other significantly differentially expressed genes that could be discussed.

--- Thank you for your kind comments. We have described more information in Discussion as follows: OsPUB57 revealed a higher expression in the resistant plants carrying the Pi9-resistant gene [48]

Line 380 Please, remove the hyphen in "elements"

--- I am grateful for your detailed comments. We revised it in Line 398 of the manuscript as follows: cis-acting elements.

Line 384 Change “transcriptiome” to “transcriptome”

--- I am grateful for your detailed comments. We revised it in Line 402 of the manuscript.

Line 388 Remove the hyphen in “patterns”

--- I am grateful for your detailed comments. We revised it in Line 403 of the manuscript.

Line 389 Change “OsPUB gene” to OsPUB genes”

--- I am grateful for your detailed comments. We revised it in Line 407 of the manuscript.

Table 2 Please, change “Gebberellin-responsive element” to “Gibberellin-responsive element”

--- I am grateful for your detailed comments. We revised it in Table 2 of the manuscript.

Table 2 Please, change “Cis-acting element involved in low-temperature resonsiveness” to “Cis-acting element involved in low-temperature responsiveness”

--- I am grateful for your detailed comments. We revised it in Table 2 of the manuscript.

Line 434, Was the quality of the RNA samples evaluated only by the NanoDrop One?

--- I am grateful for your detailed comments. We described the method of RNA sample evaluation in “Materials and Methods” (Line 460) of manuscript.

Line 436 Please, change “cDAN” in “cDNA”

--- I am grateful for your detailed comments. We revised it in Line 462 of the manuscript.

Line 438, Please, could you indicate which library preparation kit was used? and how was the quality of the cDNA libraries assessed?

--- Thank you for your critical comments. We adescribed it in “Material and Methods” (Lines 464~467) of the manuscript.

Reviewer 2 Report

Title: Genome-wide analysis of biologically important cis-regulatory elements of E3 ubiquitin ligase gene family in rice (Oryza sativa L.)

Summary: The study describes the phylogenetic, sequence, and expression analysis of 77 E3 Ub ligase genes in rice. The genes were grouped according to the sequence comparison and gene structure. Protein motifs were also identified and compared among the different groups. Expression of the genes under drought and rice blast infection was surveyed, revealing a group of 16 genes with a shared response between drought and blast infection response. The cis-regulatory elements of genes in this group were identified using PlantCARE, and their expression was verified via qRT-PCR. The authors claim that the findings here can form a basis for future work using gene editing.

Comments:

The work is flawed for several reasons, the primary one being that the title does not truly reflect what was done in the study. What was written looks more like a collection of datasets that could work together, but there is no real cohesion between them and in my opinion, does not help build the story.

  1. The title talks about several things. “Genome-wide” makes it seem that the authors searched for the genes themselves, when there already was annotations that point to these genes. “Biologically important” is too broad for what is being discussed in the study, which is really just drought and blast infection. What about other processes? Are growth and development processes unimportant? Why did the authors use a strong title for a small fraction of the “biologically important” processes in a plant? The authors also give focus on “cis-regulatory elements” in the title, yet this important piece of data comes near the end of the study. If the focus should be on how the E3 Ub genes are regulated by the cis elements they have, why is it at the end? The authors should have started with the analysis of those CREs at the beginning, rather than focusing on the gene structures of the E3 Ub genes.
  2. Table 1 could be more succinctly presented. Some of the information there could be omitted or placed in the supplementary.
  3. What is the use of the gene structure analysis? Would it not have served better for the purpose of the paper to compare the cis elements present in the 77 genes, see the if there are overlaps among them, and then compare those which are expressed in the two stresses that they had expression data for. That way, they could find the relevant cis elements for the biological processes they focus on based on elimination.
  4. Expression studies can be hard to interpret because of potential allelic differences between different genotypes. So why was Dongjin used for expression analysis instead of Nipponbare, which is the genotype sequence in RAP-DB? How confident are the authors that there are only negligible differences between Dongjin and Nipponbare? The potential issue is that the expression patterns seen with Dongjin may be because of different cis elements than those identified in Nipponbare. This could potentially cause some confusing or errant results.
  5. The authors did not state how the rice blast infection experiment was conducted in the methods section.
  6. The authors should include their primers in a supplementary table.
  7. How many replicates were used in the RNA-seq analysis? Typically, if there is enough replication for the RNA-seq, the qRT-PCR becomes redundant. Additionally, if the authors really wanted to show how correlated the values are between the two expression assays, they could have used a scatter plot. There is also a concern on the qRT-PCR results, given that only some graphs are not showing statistical significance despite having very large differences in expression between control and treatment. Does this mean that the values are very scattered between replicates?
  8. Student’s t-test was used to determine DEGs. Are there enough replications or samples for this test? Other tests are there, which are utilized by software such as EdgeR or Deseq. Additionally, there is strong skepticism on using the t-test for this purpose, especially since there may have false positives here.
  9. There are several typographical errors in different sections of the manuscript. Please review and correct them.
  10. It is a stretch to conclude that this will help in gene editing efforts. Since the authors put this at the end of the abstract and the end of the discussion, this means that this is a strong point for them. However, how exactly will this be useful? Did they identify alleles that can be edited? Statements such as this reach too much, promoting the false promise that we can solve everything by gene editing.

Overall, the authors presented a study that is not really in line with the title. There should have been a good backbone for a story here, but the way the study was done could be improved in many ways. The authors need to re-assess what they have and restructure how they want to answer the question of what are the relevant CREs in the Ub ligase genes they are looking at.

Author Response

Responses to Reviewer 2

Comments and Suggestions for Authors

Summary: The study describes the phylogenetic, sequence, and expression analysis of 77 E3 Ub ligase genes in rice. The genes were grouped according to the sequence comparison and gene structure. Protein motifs were also identified and compared among the different groups. Expression of the genes under drought and rice blast infection was surveyed, revealing a group of 16 genes with a shared response between drought and blast infection response. The cis-regulatory elements of genes in this group were identified using PlantCARE, and their expression was verified via qRT-PCR. The authors claim that the findings here can form a basis for future work using gene editing.

Comments:

The work is flawed for several reasons, the primary one being that the title does not truly reflect what was done in the study. What was written looks more like a collection of datasets that could work together, but there is no real cohesion between them and in my opinion, does not help build the story.

  1. The title talks about several things. “Genome-wide” makes it seem that the authors searched for the genes themselves, when there already was annotations that point to these genes. “Biologically important” is too broad for what is being discussed in the study, which is really just drought and blast infection. What about other processes? Are growth and development processes unimportant? Why did the authors use a strong title for a small fraction of the “biologically important” processes in a plant? The authors also give focus on “cis-regulatory elements” in the title, yet this important piece of data comes near the end of the study. If the focus should be on how the E3 Ub genes are regulated by the cis elements they have, why is it at the end? The authors should have started with the analysis of those CREs at the beginning, rather than focusing on the gene structures of the E3 Ub genes.

--- We are grateful for your critical comments. We changed the title of the manuscript as follows: We changed the title to ‘Molecular and functional analysis of U-box E3 ubiquitin ligase gene family in rice (Oryza sativa)’ based on your comments.

  1. Table 1, could be more succinctly presented. Some of the information there could be omitted or placed in the supplementary.

--- We are grateful for your critical comments. We moved Table 1 to 'Supplementary Table 1' on the information of 77 OsPUB genes as commented.

  1. What is the use of the gene structure analysis? Would it not have served better for the purpose of the paper to compare the cis elements present in the 77 genes, see the if there are overlaps among them, and then compare those which are expressed in the two stresses that they had expression data for. That way, they could find the relevant cis elements for the biological processes they focus on based on elimination.

--- We are grateful for your critical comments. In plants, PUB genes are known to have several types of domains. Therefore, we have tried to provide each gene structure of OsPUB gene family through the analysis of exon and intron for PUB genes in rice. In parallel, as cis-elements are positioned upstream of gene, we want to provide their cis-elements information in related to the gene expression for OsPUB gene family so that they could be used for future gene editing experiments to modify the gene expression.

  1. Expression studies can be hard to interpret because of potential allelic differences between different genotypes. So why was Dongjin used for expression analysis instead of Nipponbare, which is the genotype sequence in RAP-DB? How confident are the authors that there are only negligible differences between Dongjin and Nipponbare? The potential issue is that the expression patterns seen with Dongjin may be because of different cis elements than those identified in Nipponbare. This could potentially cause some confusing or errant results.

--- The reason why Dongjin was used for RNA-sequencing is that Dongjin has been used as a research material in many reports and widely grown in rice field in Korea so that we want to provide the detailed information on molecular and functional characteristics for future gene editing.

  1. The authors did not state how the rice blast infection experiment was conducted in the methods section.

--- Thank you for the detailed comments. We described experimental method about rice blast infection

  1. The authors should include their primers in a supplementary table.

--- We added the primer sets information in ‘Supplementary table 2’.

  1. How many replicates were used in the RNA-seq analysis? Typically, if there is enough replication for the RNA-seq, the qRT-PCR becomes redundant. Additionally, if the authors really wanted to show how correlated the values are between the two expression assays, they could have used a scatter plot. There is also a concern on the qRT-PCR results, given that only some graphs are not showing statistical significance despite having very large differences in expression between control and treatment. Does this mean that the values are very scattered between replicates?

--- We are grateful for your critical comments. We used three replication copies for RNA-sequencing analysis. In addition, the correlation between the two data was analyzed and attached as a Supplementary figure 1.

  1. Student’s t-test was used to determine DEGs. Are there enough replications or samples for this test? Other tests are there, which are utilized by software such as EdgeR or Deseq. Additionally, there is strong skepticism on using the t-test for this purpose, especially since there may have false positives here.

--- Thank you for the detailed comments. We used enough samples for t-test with 3 replications. And it was specified in “Materials and Methods”.

  1. There are several typographical errors in different sections of the manuscript. Please review and correct them.

--- We are grateful for your critical comments. We revised it in the manuscript

  1. It is a stretch to conclude that this will help in gene editing efforts. Since the authors put this at the end of the abstract and the end of the discussion, this means that this is a strong point for them. However, how exactly will this be useful? Did they identify alleles that can be edited? Statements such as this reach too much, promoting the false promise that we can solve everything by gene editing.

--- We are grateful for your critical comments. We revised the paragraph as follows: The molecular information of U-box E3 ubiquitin ligase gene family in rice, including gene expression patterns and cis-acting regulatory elements could be useful for future crop breeding program for producing stress-tolerant crops.
